# Spotting the Pattern: A Review on White Coat Color in the Domestic Horse

**DOI:** 10.3390/ani14030451

**Published:** 2024-01-30

**Authors:** Aiden McFadden, Micaela Vierra, Katie Martin, Samantha A. Brooks, Robin E. Everts, Christa Lafayette

**Affiliations:** 1Etalon Inc., Menlo Park, CA 94025, USA; mvierra@etalondx.com (M.V.); kmartin@etalondx.com (K.M.); reverts@etalondx.com (R.E.E.); clafayette@etalondx.com (C.L.); 2Department of Animal Sciences, UF Genetics Institute, University of Florida, Gainesville, FL 32611, USA; samantha.brooks@ufl.edu

**Keywords:** horse, mutation, coat color, depigmentation, white spotting

## Abstract

**Simple Summary:**

The understanding of coat color genetics in the domestic horse has advanced largely in the last few years. Specifically, many alleles influencing equine white spotting have been described in the last few decades. White spotting phenotypes can range from small white facial spots to an entirely white horse. Although white markings may be the most obvious phenotype, many white-causing alleles also cause adverse health defects, including sterility, deafness, and blindness. Many white spotting alleles are considered embryonic lethal in the homozygous state, causing obstacles in the breeding process. This review aims to concisely summarize recent research related to the genetics of equine white spotting phenotypes.

**Abstract:**

Traits such as shape, size, and color often influence the economic and sentimental value of a horse. Around the world, horses are bred and prized for the colors and markings that make their unique coat patterns stand out from the crowd. The underlying genetic mechanisms determining the color of a horse’s coat can vary greatly in their complexity. For example, only two genetic markers are used to determine a horse’s base coat color, whereas over 50 genetic variations have been discovered to cause white patterning in horses. Some of these white-causing mutations are benign and beautiful, while others have a notable impact on horse health. Negative effects range from slightly more innocuous defects, like deafness, to more pernicious defects, such as the lethal developmental defect incurred when a horse inherits two copies of the *Lethal White Overo* allele. In this review, we explore, in detail, the etiology of white spotting and its overall effect on the domestic horse to Spot the Pattern of these beautiful (and sometimes dangerous) white mutations.

## 1. Introduction

A streak of white, a patch of pink skin, a piercing blue eye. These traits add individuality, stunning beauty, and economic value to the domestic horse (*Equus caballus*). Specific color traits can help a horse qualify for color-specific registries. However, some of these prized alleles can cause detrimental phenotypes, such as an increased risk of deafness or blindness. Other combinations of these sought-after color traits fail to produce viable offspring. Understanding the etiology of white markings is crucial to ensure ethical care and breeding practices. 

Recent advancements have decreased the cost of genome sequencing, enabling the elucidation of the characteristics of the genetic mutations causing many depigmentation phenotypes. *KIT*, *MITF*, *PAX3*, *HPS5*, *EDNRB*, *TRPM1*, and *RFWD3* represent the collection of genes associated with different white markings in the domestic horse (Table 1). While the genetic mechanisms underlying some of these color traits are not fully understood, the information collected on white spotting mutations can assist breeders in making optimal choices to breed unique and healthy herds. In this review, we explore the mechanisms and phenotypic effects of equine depigmentation and their overall effect on horse health. 

## 2. Dominant White

A “white” horse has held a special place throughout history and mythology, appearing in many great legends and tales. However, the horses captured in the literature and art do not always depict a truly white horse, whose phenotypes are often confused with *Grey* or *Cream*. Because horses homozygous for *Cream* have pink skin and hair diluted to a near-white color, it is difficult to differentiate between true white and homozygous *Cream* horses [1]. *Dominant White* horses are born with white markings and display pink skin below these areas, while individuals with *Grey* do not have pink skin and are not born with white hairs, but develop them with age [1]. The Romans knew of the phenotypic differences between gray and white, although it remains unknown if their terms for these colors correspond to modern designations [2]. Investigations into runs of homozygosity in 1476 horses of European descent revealed positive selection for base coat color on ECA3, but not in the region harboring *KIT* [3], while selection for white coat color patterns has been identified in ancient horse DNA [4]. 

The *Dominant White* locus is the equine locus with the largest number of known variants causing depigmentation (Table 2), and it is located on chromosome 3 within the *Proto-Oncogene*, *Receptor Tyrosine Kinase* (*KIT*) gene [2]. *Dominant White* uses the capital “*W*” followed by the integer in the series to indicate the specific dominant variant present in the genotype (e.g., *W35*). Originally, the *W* symbol was used for a small number of variants all following a true dominant pattern of inheritance and producing all white horses in the heterozygous state. The earlier dominant mutations were not observed in the homozygous state, leading to the adoption of an alternate term, *Lethal Dominant White*. Time has not honored this tradition as, to date, six *W* variants are known to be inherited in an incomplete dominant manner, with genotypes existing in the homozygous state in apparently healthy horses. Recent publications have started to refer to *Dominant White* as *White Spotting* to better account for the varied phenotypes at the *W* locus [5,6,7,8]. Phenotypes at the *Dominant White* locus are broadly characterized by horses displaying white areas with clear borders, or a completely white horse with pink skin underneath. Thirty-five *W* variants have been reported, and it seems likely that the number will continue to increase [2,9,10,11,12,13,14,15,16,17,18,19,20].

Many *W* alleles have been traced to a founding individual and are limited to those descendants, yet others are observed in diverse breeds, including a few that have likely circulated among diverse breeds over centuries, transmitted in cross-breed matings and by shipment of horses around the globe. The introduction of white alleles to new breeds is also promoted by registries opening studbooks to foreign horses with hopes of reducing inbreeding. As an example, researchers identified *W13* in American Quarter Horses (AQH) in 2011 [14], but more recently observed this allele in both Shetland ponies and American Miniatures [21], two breeds genetically distant from the AQH. As Shetland ponies are not typically tested for *W13*, identification of this allele only occurred after genotypes for more common white variants (ex: *TO*, *SB1*, *W20*) failed to explain the depigmentation. Identifying variants outside of their presumed breed of origin is important to ensure the accurate monitoring and reporting of alleles with harmful side effects. Increased awareness of the presence of these alleles in new and existing populations will help prevent the introduction of white alleles into registries that select against white markings and mitigate the potential crossing of lethal pairs. 

### 2.1. Phenotype

Phenotypes associated with the *W* locus are characterized by either white patterning or an entirely white coat with pink skin underneath. Mild white spotting phenotypes are described as sabino-like, with white legs, facial stripes, and a collection of other white facial markings (called stars or snips depending on the size, shape, and location), and, less commonly, patches of white hair across the abdomen [22]. Strongly deleterious mutations (frameshift, stop-gain, indel) typically result in completely white horses with pink skin (Figure 1). In contrast, *W20*, *W32*, *W34*, and *W35* horses can be solid (non-white) in color in the absence of other white alleles but may magnify white markings caused by other alleles. For example, when an individual carries one copy of *W22* and an out-of-phase copy of *W20*, the resulting phenotype is an all-white or almost all-white horse, despite each of these individual alleles typically producing less pronounced depigmentation phenotypes [11,23]. The amplification of the degree of white spotting is also observed with *W5/n* and *W20/n* compound heterozygotes, as these horses display an all-white phenotype [24]. Variants *W19*, *W21–W23*, *W28*, and *W31–W35* produce a sabino-like phenotype sometimes accompanied by depigmentation on the abdomen with jagged borders. *KIT* variants also sometimes cause a rare phenotype of blue eyes when the depigmentation covers the entire face, including the eyes.

Deleterious *Dominant White* alleles result in more extensive white markings and are likely lethal in the homozygous state [25]. *W1–W14*, *W16–W18*, *W21–28*, *W30*, *W31*, and *W33* have not been observed in the homozygous state, and are predicted to be homozygous lethal due to their similarities to mutations observed in other species [2,25]. Progeny ratios for white alleles causing fully white phenotypes also stray from Mendelian expectations. When heterozygous white horses were crossed, the resulting offspring possessed a 2:1 ratio of white foals to solid foals, supporting the hypothesis that *W/W* is lethal during early gestation [25]. These two observations suggest homozygous embryos are not viable for certain alleles, but too few births have occurred to conclusively determine the lethality of each variant. *W15*, originally thought to be embryonic lethal, was later reported in two homozygous individuals [18]. A horse homozygous for *W19* was also recently identified, which also boasted two copies of *W34* and *W35* each, for a total of six white spotting variants [26]. Cases such as this support the hypothesis that other white variants could be viable in the homozygous state but have not yet been observed. Conclusions regarding the lethality of homozygous *Dominant White* variants will only be elucidated through continued monitoring and expanded genetic testing for these variants. 

Many *KIT* variant haplotypes are reported in horses, and the resulting phenotypes are extremely varied and not fully documented. Phenotypes of horses with multiple white alleles depend on the specific white allele combination but generally result in increased depigmentation when compared to individuals with only one variant. Despite the impressive number of publications on the *Dominant White* locus, there are few studies focusing on phenotypes of horses with multiple white alleles. There are even fewer studies focusing on the health effects, and specifically, the reproductive effects, of horses with *KIT* variants, despite reports of *KIT* variants being associated with health defects in other species [2].

### 2.2. Mechanisms and Genetics 

KIT transmits transmembrane signals critical for survival and plays an important role in melanogenesis [27,28,29]. During development, melanoblasts begin to migrate from the neural crest to populate the rest of the body and eventually develop into melanocytes (pigment-producing cells). Melanocyte development is in part controlled by interactions between KIT, KIT Ligand (KITL), and Melanocyte Inducing Transcription Factor (MITF) [28,29,30,31,32,33,34,35]. After binding with KITL in the extracellular domain, KIT self-dimerizes and phosphorylates MITF, activating the transcription factor and upregulating target genes involved in pigmentation [28,29]. Mutations affecting the function or binding sites of KIT protein disrupt this pathway, resulting in downregulated pigment genes and melanocytes failing to develop in some or all of the tissues. 

There are a variety of mutations at the *Dominant White* locus including deletions, insertions, missense, nonsense, and splice site variants. More impactful mutations alter the protein conformation and function to a greater degree, and cause greater disruptions to KIT pathways, resulting in fewer melanoblasts properly migrating and a more depigmented individual. More tolerated *KIT* variants such as *W20*, *W32*, and *W35* have subtle effects on the protein or protein expression and result in milder phenotypes. However, because the failure of a melanoblast to migrate is a chance event, mild variants on their own may still cause extensive depigmentation. The stochastic nature of white spotting events can cause individuals with the same genotype to display very different phenotypes. Commercial genetic testing for all *W* alleles exists, but assays for *W10*, *W13*, *W19*, *W20*, and *W22* are among the more readily available tests since these alleles are more common.

Up to three *KIT* variants have recently been found linked together, resulting in complex haplotypes. To date, the *W22* allele has only been observed in combination with the *W20* allele [11,23]. *W19* has been observed by itself and in linkage with *W34* and *W35*. The *W19W34W35* haplotype likely occurred by a crossover event because it was only identified in an inbred family, while the *W19* allele has been found out of phase of *W34* and *W35* in multiple families [26]. Sixteen haplotypes, including combinations of *W20*, *W32*, *W34*, and/or *W35* with other variants, have been identified, but the genesis of these complex haplotypes is not completely understood. Founder horses have not been reanalyzed for recently discovered alleles to reveal if novel variants occurred on the background of other alleles or if the haplotype occurred via a crossover event. While phenotypic records of all known multilocus genotypes are incomplete, it is likely that more white variants increase the amount of white patterning on a horse.

## 3. Tobiano, Sabino, and Roan

*Tobiano*, *Sabino*, and *Roan* are three alleles that have adopted the name of the respective phenotype. All three were independently mapped to ECA3 near the *KIT* gene [9,27,36,37,38,39]. *Tobiano* and *Sabino* are two well-characterized variants whose origins trace back to ancient times. Conversely, *Roan* is poorly understood, and no causative variant has been identified to date, though it has been mapped to the *KIT* region. All three variants cause varying degrees of white phenotypes in the heterozygous state without disrupting the base coat color. Homozygous *Sabino 1* horses are mostly all white, while *Tobiano* homozygotes display varied amounts of depigmentation. Although roaning can be localized to a region or cover the entire body, the *Roan* allele is specifically characterized by roaning covering the body of the horse, leaving the head and extremities dark. The genomic coordinates of these three variants are shown in Table 3.

### 3.1. Tobiano 

The nomenclature for *Tobiano* uses “*TO*” to represent the presence of the dominant allele and “*to*” to indicate the wild type allele. The mutation causing the tobiano phenotype, characterized by large white areas with smooth borders on the body and legs, has been mapped just outside of the *KIT* gene [1,36]. The oldest horse identified with the *Tobiano* allele dates between 3500 and 3000 years ago in Eastern Europe. The frequency of the *TO* allele has changed throughout history. During early domestication, the frequency of *Tobiano* began to rise, but it fell during the Middle Ages, as humans selectively bred in favor of other coats [4,40]. 

#### 3.1.1. Phenotype

Classic tobiano, minimal tobiano, and cryptic tobiano are terms used to describe the diversity of phenotypes displayed by individuals with the *Tobiano* allele (Figure 2). Classic tobiano horses display large patches of white spotting, with clear borders [1,24,36,41]. Classic markings often spread over the flank, stretch up the neck, and cross the back. Classic tobiano horses generally have four white legs below the knees, but these marks can extend higher. Minimal tobiano, as the name suggests, is characterized by fewer white markings on the body, similar in pattern to classic tobiano but smaller in distribution. Cryptic tobiano horses display little to no white markings on the body, a phenotype easily confused for solid horses with white socks [41]. This presents a challenge in breeding when genotypes are unknown and has led to inaccurate heredity reporting in studbooks [24]. Cryptic tobiano individuals mistaken for solid horses with socks can also introduce the *Tobiano* variant into studbooks where the trait should be excluded [41]. Tobiano horses express their base coat color (chestnut, black, bay) in pigmented areas and have black skin beneath; however, the skin beneath white areas is pink. Individuals displaying tobiano characteristics often have a mix of white and base coat-colored tails. This phenotype is unique to *Tobiano* and is not expected for any other known white spotting trait [1,36]. Homozygous *Tobiano* horses may display “ink spots,” which are small dark spots within white areas, but the *TO/TO* genotype does not explain all cases of ink spots [1]. *Tobiano* horses with other white spotting variants often express an additive phenotypic effect and are more white than horses with *Tobiano* alone.

#### 3.1.2. Genetics and Mechanism

The mutation causing the tobiano phenotype is a large paracentric inversion of a ~43 Mb region upstream of *KIT* and is expressed in a dominant manner [1,36]. This region, spanning nearly a third of the chromosome, is thought to harbor genetic machinery regulating *KIT* and its relocation is believed to reduce *KIT* expression. The KIT structure is not directly affected by this inversion, allowing for proper protein function, which may explain the possibility of the cryptic tobiano phenotype. Many homozygous *TO/TO* individuals have been documented and none have been reported to have any health defects due to the *TO* allele [1,36]. The *TO* mutation may result in reduced fertility rates, as large chromosomal inversions typically make gametogenesis more difficult, but this has not been studied [36]. The *Tobiano* allele is found in diverse registries including but not limited to: Paint, Pinto, Shetland, Miniature, Warmblood, Gypsy, and Paso Fino. Conversely, the trait is not commonly found in American Quarter Horses, Thoroughbreds, Standardbreds, or Arabians. Commercial genetic tests for *Tobiano* are readily available.

### 3.2. Sabino

The sabino phenotype is the earliest known white spotting phenotype and is one of the best characterized white spotting mutations. The genetic variant responsible for *Sabino 1* traces back to sometime between 5500 and 5000 years ago, first appearing during early domestication on the Siberian Steppe and later found in Armenia and Moldavia [4,40]. The sabino trait uses the symbol “*SB1*” to represent the presence of *Sabino* and “*sb1*” or “*n*” to indicate the absence of *Sabino*. The phenotype produced by the *Sabino* mutation manifests in a manner similar to *Dominant White* mutations like *W19* and *W31*. These visual similarities can cause confusion between *Sabino* and other white-causing traits. Genetic testing is the best way, and sometimes the only way, to differentiate a true *Sabino* from a *Dominant White* horse. 

#### 3.2.1. Phenotype

The sabino phenotype is characterized by extensive depigmentation of the legs, face, and abdomen [1,9]. Although the sabino phenotype is attributed to the *Sabino 1* allele, *Dominant White* variants can also produce sabino-like phenotypes, making distinctions between sabino phenotypes and other white spotting phenotypes challenging. Heterozygous *Sabino* horses display irregular, jagged borders to their white markings, but homozygous horses are fully or nearly fully white (Figure 3). *SB1/n* individuals may have white speckles near the borders of larger areas but they are typically surrounded by pigmented areas with clear borders [1,9]. Pink skin is found below the white areas and on the face, while skin below pigmented hair is normally pigmented. Homozygous individuals and heterozygotes with extensive white spotting are more sensitive to light and should be monitored closely when exposed to direct sunlight for prolonged periods. Horses with *SB1* may express a minimal sabino phenotype with low white socks similar to cryptic tobiano. Horses with *SB1/n* and other white spotting variants typically express more white than horses with *SB1/n* alone.

#### 3.2.2. Genetics and Mechanism

The *Sabino* phenotype is inherited in an incompletely dominant manner and is caused by a splice site SNP in *KIT* intron 16 [1,9]. This splice site variant produces a unique transcript lacking the 17th exon, which encodes part of the active kinase domain. Without this critical exon, KIT may fold in a less effective manner or be completely inactive. Having less active KIT results in fewer melanoblast signaling events and less developed melanocytes, resulting in the absence of pigment with a white spotting or all white phenotype, depending on zygosity. However, because some copies of the normal transcript are still expressed, heterozygous and homozygous horses develop into healthy individuals with no reported health defects [1,9]. Genetic tests for the sabino mutation are readily available.

*SB1* is an old mutation and has had many generations to diffuse through breeds and combine with other white spotting mutations [1,4,9,40]. Haplotypes containing the *SB1* allele and the *W20* and/or *W32* have been identified [26]. The *Sabino* mutation is found in a diverse set of breeds, many of which are of Spanish origin. Although a sabino-type phenotype is found in Clydesdale horses, the *SB1* variant is absent, indicating multiple pathways lead to phenocopies. 

### 3.3. Roan

Although the phenotype caused by the *Roan* allele is well characterized, the genetic cause remains a mystery. The symbol “*RN*” is used to indicate the presence of the allele and “*rn*” or “*n*” to indicate its absence [1]. The roan phenotype is characterized by a mixture of white hairs diffused throughout the body, while the head, lower limbs, skin, mane, and tail remain solid in color (Figure 4). “Blue roan” is used to describe the roan phenotype for horses with a black base coat color and “red roan” is reserved to describe the roan phenotype on a chestnut coat. The phenotype associated with the *Roan* allele can often be confused with varnish roan, which is attributed to the *LP* allele. The *Grey* phenotype may also be confused for *Roan*, with the most notable difference being that the *Grey* trait progressively lightens the coat with age, whereas *Roan* individuals are born with white ticked coats. So far, only anecdotal evidence exists for homozygous *Roan* horses, since the trait does not yet have a well-documented mode of inheritance or a known causative variant. Pedigree analysis suggests that *Roan* is inherited in a dominant manner [37,38,39].

Multiple studies observed the association of *Roan* to markers within the *KIT* gene on ECA3, but there is no clear consensus regarding any single SNP most strongly associated with the roaning phenotype. *KIT* is the gene most likely to harbor the variant causing roan but a causal mutation has not been identified [39]. They excluded an intronic insertion of a LINE as the causative mutation after discovering it was common in all horses. The team also identified a synonymous mutation in exon 19 strongly associated with the roaning phenotype in all breeds except Shetland and Gotland ponies [39]. A genetic marker in the 17th intron of *KIT* was found to be the most associated with the *Roan* phenotype [37]. The proposed genetic marker was completely associated with *Roan* in Noriker horses and is suggestively associated in Quarter horses and various draught breeds [37]. Yet, this marker fails to explain the roaning phenotype in all breeds, as Shetland ponies with the *Roan* phenotype did not possess the associated marker. The breed segregation of these discoveries and conflicting conclusions regarding lethal *Roan* suggests there may be allelic heterogeneity for *Roan* or the presence of multiple phenocopies.

Roaning in canines is caused by a tandem intronic duplication in the *USH2A* gene [42], and in mice by a deletion in the mouse ortholog of *KITL*, the *Steel Ligand Factor* (*SLF*) gene, encoded by the *Steel* locus (reviewed in 60). These reports suggest the *Roan* allele in horses may not be exclusively controlled by mutations in *KIT*, but neither of these comparative candidate genes lies on ECA3 and would therefore not likely be involved in the *Roan* phenotypes within breeds where an association with ECA3 was observed. If the roan phenotype is caused by multiple variants, or by epistatic interactions between multiple loci, many of the existing disagreements about *Roan* could be explained. Top-associated SNPs differ between breeds and studies, bringing into question the validity of currently marketed *Roan* assays, as these are easily applied inappropriately to populations in which the association of these markers to the *Roan* phenotype is unknown.

## 4. Splashed White

The *Splashed White* collection is the second largest grouping of genetic variants causing depigmentation in horses [15,43,44,45,46,47,48]. The capital letters “*SW*” followed by the number in the series (e.g., *SW10*) is used to indicate a mutant allele, with an “*n*” or “+” used to indicate the wild-type allele [1]. There are ten reported variants in the *Splashed White* grouping, each affecting one of two transcription factors: *PAX3* (*Paired-Box 3*) or *MITF*. Although these two genes are completely independent loci, and play different roles in the cell, mutations in either gene causes phenotypes without distinguishable differences, leading both to be included in the *Splashed White* naming convention (Table 4). The *Splashed White* trait was named after the resulting phenotype, as affected horses feature patches of white that make it look as if they have been dipped in or “splashed” from below with white paint. White markings typically exist on all four legs extending up to and sometimes covering the abdomen and the face. 

### 4.1. Phenotype

As mentioned above, the phenotype characterizing *Splashed White* features flashy white legs with additional white markings on the abdomen that can extend all the way up to the body. Some variants produce fully white abdomens, with smooth borders and little to no roaning or speckling, as if the horse dipped its belly in white paint [1]. These horses often also demonstrate all white, “paint dipped” faces: a facial phenotype sometimes referred to as “bald.” In horses with *Splashed White*, markings can extend beyond the eyes and ears (Figure 5), accompanied by blue or partially blue eye(s). There are no major differences between *Splashed White* phenotypes caused by mutations in *MITF* and *PAX3*, and mutant alleles for either gene are tied to an increased risk of deafness [43,44,45,46,48]. Depigmented individuals may be more sensitive to sunlight and require more care when in direct sunlight for longer periods. Combinations of *SW* variants are less frequent than combinations of *Dominant White* alleles, so less is known about their genetic interplay and resulting phenotypes. However, in some observed cases where at least one *MITF SW* variant and at least one *PAX3 SW* variant are inherited (most commonly, *SW1/n SW2/n*), or two *MITF SW* variants are inherited (*SW1/n SW7/n*), horses can be born with an all-white or nearly all-white coat and can be deaf depending on the variants inherited [44,48].

### 4.2. MITF

*MITF* encodes a transcription factor necessary for proper cell development [29,30,31,32,49]. MITF contains one basic helix-loop-helix and one leucine zipper domain, providing specificity to and affinity for its DNA target sites [35,50]. MITF is phosphorylated by KIT during melanogenesis to direct the gene expression pathways that differentiate melanoblasts into melanocytes and later induce the expression of other pigment genes such as *TYR*, *TYRP1*, *PMEL*, and *MLANA* [34]. Similar to other mutations found within DNA binding domains, the change in protein structure causes reduced affinity or specificity in binding, causing MITF to stray from its typical targets and pathways.

Seven of the ten *Splashed White* alleles are *MITF* mutations on ECA16 [43,44,45,46,47,48]. Most of these alleles are inherited in a dominant manner, meaning individuals with one mutant allele typically demonstrate a *Splashed White* phenotype. For most of these mutations, homozygous individuals have not yet been observed, and this, in combination with the dominant inheritance pattern, implies that these mutations could be embryonic homozygous lethal. To date, the only *MITF* allele observed in the homozygous state is *SW1*, which is inherited in an incomplete dominant manner. Homozygous individuals display typical splashed markings, but heterozygotes display less extensive depigmentation [1]. *SW1/SW3*, *SW1/SW2*, *SW1/SW5*, and *SW1/SW7* horses have been observed and display all white phenotypes with signs of deafness [44,48]. No homozygous individuals have been described for *MITF Splashed White* mutations outside of *SW1*. Most *MITF* variants are frameshift indels with significant impact and likely lethal in the homozygous state, which may explain why homozygous individuals are not observed. While *SW1* is found in numerous breeds, the remaining *MITF* alleles are breed specific. *SW9* is only found in a single family of Pura Raza Española horses, while *SW3*, *SW5*, *SW6*, and *SW7* have only ever been reported in Paint and Quarter horses [44,45,46,48]. *SW8* was found in a single Thoroughbred stallion and one of his offspring [43]. Parentage testing confirmed the Thoroughbred proband as the *SW8* founder. 

There are three *MITF* mutations associated with depigmentation without an *SW* designation [44,51]. *Macchiato* is a variant of *MITF* (ECA16:21564980T>C) found in one sterile male displaying a light coffee color coat and white spotting similar to phenotypes attributed to *Splashed White* alleles [44]. The term *Macchiato* was assigned to this trait because the coat color dilution affects the base coat, while *SW* does not. A de novo *MITF* variant predicted to alter protein sequence (ECA16:21556522C>T) was identified in a single white American Standardbred [51]. Because neither mutation was likely to be passed onto future generations, they were not given an *SW* designation. An intronic variant (ECA16:21608936C>T) was associated with forelimb white markings in Menorca Purebred horses and facial white markings in Pura Raza Española horses [52]. The intronic variant is located 29.9 kb downstream from the transcription start site and is speculated to either affect the regulation of *MITF* or is linked to another variant responsible for causing white markings [52]. Although this variant was not given an SW designation, it is still a useful marker to select for white coat color in Spanish breeds. The linkage of the ECA16:21608936C>T variant to white markings has not been studied in other breeds.

### 4.3. PAX3

*PAX3* belongs to the Paired Box family of transcription factors characterized by a paired-type homeodomain and paired box domain [50,53,54]. Transcription factors in the PAX family are well understood to regulate transcription during development. PAX proteins contain an octapeptide sequence, which helps maintain DNA–protein interactions and conserve binding domains [29]. The paired box domain, as the name would imply, is the most conserved domain in the PAX family. Mutations in *PAX* genes are associated with hypopigmentation and deafness in horses and developmental abnormalities in humans and flies [1,54]. In horses, mutations in *PAX3* are thought to interrupt the encoded transcription factor’s affinity and/or specificity to bind to *MITF*. Without proper PAX3 interaction, MITF is under expressed, causing down-regulation of its target pigment proteins and eventual failure of melanoblast proliferation and differentiation [30,49,55]. The regulatory interactions between PAX3 and MITF partly explains how mutations in either gene could result in a splashed white phenotype.

Three of the ten *Splashed White* alleles are attributed to mutations in *PAX3* on ECA6 and cause depigmentation of the hair, skin, and eyes as well as an increased risk for deafness [15,44,47]. Of the *Splashed White PAX3* variants, only *SW2* has been observed in the homozygous state, resulting in all white phenotypes. *SW2* is inherited in an incomplete dominant manner with heterozygous individuals displaying typical *Splashed White* markings. *SW4* is predicted to be nonviable in the homozygous state due to a lack of observed homozygotes. *SW10* is also predicted to be embryonic lethal in the homozygous state since it terminates the transcript before a crucial DNA binding domain via an early stop-gain missense mutation. However, too few births have occurred to definitively conclude lethality in both cases. *SW2* and *SW4* have been observed in Quarter horses and Appaloosas, respectively [15,44]. To date, *SW10* has been observed in two Pura Raza Española horses [47]. Interestingly, the families with either *SW9* or *SW10* have been bred together, making fertilization of a *SW9/n SW10/n* zygote possible, although no individuals with this genotype have been observed as of yet, if even viable. It is unclear whether combinations of *PAX3* mutations are viable, as all alleles exist in low frequency.

## 5. Eden White and HPS5

*Eden White* is a recently reported grouping of white spotting variants associated with *Hermansky-Pudlak Syndrome 5* (*HPS5*) on ECA7 [56]. Relatively little is known about the function of this encoding protein, but mutations in *HPS5* cause depigmentation in humans, mice, and other organisms [56]. *HPS5* encodes a member of Biogenesis of Lysosome Related Organelles Complex 2 (BLOC-2) with members HPS3 and HPS6, but the exact function of the complex is unknown. It is therefore unknown how mutations in *HPS5* affect the complex’s function and result in depigmentation. HPS5 has one annotated domain, WD40, thought to control interprotein interactions in BLOC-2 [56]. Although the exact function of the BLOC-2 complex is unknown, it is suspected to play a role in organelle biogenesis associated with melanosomes, lysosomes, and platelet-dense granules [56]. In humans, mutations in *HPS5* cause Hermansky–Pudlak syndrome type 5, characterized by depigmentation and a reduction of platelet-dense granules, resulting in prolonged bleeding times [56]. Mutations in *HPS5* have also been reported to cause albinism in zebrafish [56].

Three variants within *HPS5*, termed *Eden White 1* through *Eden White 3 (EDXW1-EDXW3)*, have been associated with depigmentation in horses (Table 5). The phenotypes of affected horses are similar to *Splashed White* phenotypes, featuring sabino-like markings, long blazes, and socks with distinct borders (Figure 6). *EDXW* variants may also produce white marks on the abdomen and blue eyes (Figure 6A), but these are more rare phenotypes, and are usually only found among homozygous or compound heterozygous individuals. The symbol *EDXW* followed by the number in the series (e.g., *EDXW2*) is used for the mutant while the letter “*n*” is used to indicate the absence of any *EDXW* variant. Horses of all major breed groups have been found to possess *Eden White* alleles, but heavy horses and stock-type horses are most commonly found with these variants. To date, no health defects have been reported for horses with any of the *Eden White* variants, but this has not been thoroughly investigated. 

Although it is possible to have combinations of *Eden White* variants, including compound homozygous, there are no reported trends for the phenotypes of these horses [56]. It has been observed that they display more white spotting but too few individuals have been phenotyped to test this hypothesis statistically. It is likely that the more variants possessed, the more white markings will be displayed. *Eden White 1* and *Eden White 3* are epistatic to *MC1R*, which controls the chestnut or black coat color in horses, similar to most mammals. Individuals with a dominant black base coat color (*E/e* or *E/E*) display more white markings than individuals with a chestnut coat color (*e/e*), a trend not yet documented for *EDXW2*, but likely consistent among all non-tolerated *HPS5* variants, similar to findings in other species [56].

## 6. Lethal White Overo

Lethal White Overo, commonly known as Frame Overo, is the phenotype displayed by heterozygotes possessing the *LWO* variant. This allele is symbolized by “*LWO*” or “*O*” while “*n*” or “*o*” are used to indicate the absence of the mutation; however, other texts have used other symbols, including [1]. Horses displaying frame overo can have strong facial markings similar to *Splashed White* individuals, with their face appearing to be bald or dipped in paint (Figure 7) [57]. *LWO* horses display irregular framed depigmented patches on the body, but the markings typically do not extend below the body or cross the spine. Horses with *LWO* may have blue or partially blue eyes and may be predominantly white or their base coat color with minimal or no white markings in the heterozygous state [57]. In one population, 18% of solid breeding stock horses were heterozygous for the *LWO* allele, indicating that the frame overo phenotype is not always apparent in *LWO* heterozygotes and other factors might influence the extent of white spotting [57]. Horses born with two copies of *LWO* are entirely white and with a megacolon, which prompts immediate euthanasia [1,57]. The mutant allele is most common in American Paint Horses and proof of *LWO* genotype is one of two alleles acceptable for registry in the American Paint Horse Association [57]. Individuals with *LWO* and other white spotting mutations typically display larger white spots than individuals with only one copy of *LWO* [1]. The variable phenotypes of *LWO* and other white spotting mutations has led to the misclassification of heterozygous *LWO* horses as splashed white, tobiano, sabino, or their phenotypic derivatives [e.g., tovero, calico, and frame blend] [57].

A double nucleotide polymorphism (chr17:50503041-50503042delinsCT) in the first exon of *Endothelin Receptor B (EDNRB)* on ECA17 is responsible for *LWO*. *EDNRB* encodes a g-coupled receptor responsible for the migration of neural crest cells during early development. In embryos, the protein helps neural crest-derived cells develop into melanocytes, as well as nerves within the intestines [1,58]. The dinucleotide polymorphism occurs at codon 118, within a transmembrane domain, and exchanges a nonpolar isoleucine to positively charged lysine [58]. The exchange of the evolutionarily conserved isoleucine is thought to affect EDNRB morphology, affecting how it folds or how it anchors itself in the lipid bilayer [58]. LWO syndrome in horses is synonymous with the Hirschsprung and Waardenburg Syndromes in humans, which are similarly attributed to mutations in human *EDNRB* [57,58]. These diseases are characterized by developmental issues and depigmented eyes, hair, and skin. Humans, mice, and horses with *EDNRB*-associated diseases all exhibit similar phenotypes.

The phenotypes of homozygous *LWO* horses are observable from birth. The most apparent and immediate are the fully white coat and blue eyes, but the adverse health effects are observable soon after. Most homozygous foals show signs of intestinal discomfort within several hours of birth. These foals usually do not pass meconium, and neither surgery nor medication help to bypass the occlusion [1,58]. Affected foals are missing intestinal ganglia that control the peristaltic muscles responsible for peristalsis throughout the intestines, meaning they cannot pass food through their digestive tract. However, not every all-white foal born is homozygous for *LWO*, nor does every horse heterozygous for *LWO* display depigmentation phenotypes. For these two reasons, readily available commercial genetic testing for *LWO* is the most effective method for determining the *Frame Overo* genotype and limiting instances of breeding that could lead to lethal white syndrome-affected foals.

## 7. Leopard Complex Spotting 

*Leopard Complex* spotting, commonly referred to by the breed most known for this color, the “Appaloosa,” is an incompletely dominant phenotype caused by a mutation in the *Transient Receptor Potential Cation Channel Subfamily Member 1 (TRPM1)* gene. The symbol “*LP*” is used for the incompletely dominant allele and lowercase “*lp*” is used to indicate the recessive wild type allele. The oldest genetic evidence of leopard spotting could be traced to the Pleistocene, but negative selection for the trait post-domestication reduced its frequency [4]. The protein encoded by *TRPM1* mediates the flux of ions across membranes in the brain, heart, and melanocytes to modulate cell polarization and is thought to play a role in melanin synthesis [59]. TRPM1 is partly responsible for vision in low-light settings as visual signals from rod centers trigger the protein channel to close [59]. *Leopard Complex* spotting is caused by a retroviral insertion found within *TRPM1* (Table 6) that interrupts protein function [1,27]. The mutant allele contains a premature poly-adenylation signal, causing the expression of a non-functional *TRPM1* isoform. The resulting protein lacks the ability to respond to regulatory signals normally controlled by glutamate. Without this response, TRPM1 cannot fulfill its normal functions, severely impacting pathways including sight and pigment. The hypothesized mechanism by which *TRPM1* variants might cause albinism suggests that the mutant transcript causes morphological defects in melanocytes, which causes melanocyte death [27]. 

Horses with an *LP* allele typically display a depigmented hind quarter with pigmented spots (Figure 8), but can display more pronounced depigmentation phenotypes extending to beyond the hind quarters onto the body and neck or very minimal phenotypes of roan-like patterning or solid with little to no white markings at all. The *LP* phenotype of a white hind quarter with pigmented spots is commonly referred to as a blanket, due to its similarity to a polka-dotted blanket draped over a horse. The extent of depigmentation due to the *LP* allele is highly variable, including other phenotypic characteristics of visible white sclera, striped hooves, mottling (depigmentation/pink skin around the face, genitalia and anus), and varnish roaning (a gradual loss of pigmentation throughout the coat but not on bony surfaces) [1,27]. Horses with *LP/LP* generally display much more white and fewer spots than heterozygotes. White markings for *LP* do not generally produce the clear borders observed for other white spotting mutations. White areas can appear in patches with jagged borders and have white hairs dispersed in the surrounding area. The extent of white spotting is influenced by other factors, including sex, the *Extension* genotype, the *PATN1* genotype, and the accumulation of mutations in either *KIT* or *MITF* [59]. Although *LP* is the signature phenotype of the Appaloosa horse breed, the variant is found in several other horse breeds, including but not limited to, Knabbstrupper, Ponies of America, Paint, and Quarter Horses [1].

### 7.1. Congenital Stationary Night Blindness

The phenotypic health effect of the retroviral insertion responsible for *LP* is only observed in homozygous individuals, and the *LP/LP* genotype is the most common cause of Congenital Stationary Night Blindness (CSNB) in horses [1,27]. It is thought that the mutant TRPM1 lacks the necessary machinery to properly respond to ocular signals produced in low light levels [27]. This null response causes homozygous individuals to be born with a lack of vision in the dark that does not progress with age. Although horses can adapt well to CSNB, they are more prone to injury during the night if sufficient light is not provided. CSNB can be difficult to diagnose based on a highly variable phenotype and because horses can be well adapted to the condition. For these reasons, genetic testing is the most recommended method to verify an individual’s *LP* genotype and a possible diagnosis of CSNB.

### 7.2. Equine Recurrent Uveitis

Equine recurrent uveitis (ERU) is an ocular disease associated with *LP* in horses that may cause the development of cataracts, glaucoma, and blindness [60,61]. ERU differs from CSNB in multiple characteristics: 1. ERU usually worsens with age, 2. Horses are not born with symptoms of ERU, and 3. ERU is triggered by environmental and genetic factors as well as age [60,61]. The genetic component of ERU was associated with *LP* in multiple studies, but the association does not explain all cases of ERU [60,61]. There are likely other undiscovered traits or environmental factors also contributing to ERU risk. Horses with the *LP/LP* genotype are at a higher risk for developing ERU than heterozygotes. Due to the high frequency of *LP* in Appaloosa horses, they are eight times more likely to develop ERU than other breeds [60]. It has been predicted that periocular depigmentation might underlie the higher risk for ERU despite findings indicating varnish roan phenotype is not a significant predictor [60]. Commercial genetic testing for *LP* is available for determining an individual’s ERU risk, although the presence of this allele does not guarantee a horse will develop ERU.

### 7.3. RFWD3 and PATN1

The *LP* allele is epistatically influenced by the *Pattern-1* allele, located within the *Ring Finger and WD Repeat Domain 3 (RFWD3)* gene on ECA3 [1,59]. *PATN1* is the symbol used to represent the dominant non-reference allele and “*n*” *or* “*patn1*” is used to indicate the absence of the variant. *PATN1* is inherited in a dominant fashion as horses express the *PATN1* phenotype in the presence of at least one *LP* allele regardless of *PATN1* zygosity (Figure 8). The *PATN1* phenotype has only been observed in *LP/X* horses (where *X* is either allele) and horses without *LP* appear the same regardless of the *PATN1* genotype. Horses with *LP/lp* and *PATN1/X* display more *LP*-associated white markings while *LP/LP PATN/X* individuals are generally born nearly all white and with fewer spots. *PATN1* is associated with an increased risk of ERU in individuals with at least one *LP* allele. Individuals with *LP/LP PATN1/n* are at the greatest risk of developing ERU [60,61]. More studies are needed to explain how *PATN1* and ERU are related.

*RFWD3* encodes an enzyme responsible for ubiquitin-protein ligase activity and is involved in several pathways including cell cycle regulation and DNA repair. In response to DNA damage, the encoding protein is recruited to form a complex with Mdm2, another ubiquitin ligase. A SNP mutation in the 3′UTR of *RFWD3* is strongly associated with the *PATN-1* phenotype and is suspected to alter the protein’s expression. RFWD3 is thought to be involved in the removal of deformed melanosomes observed in *LP* horses, either directly or indirectly. Alterations in its expression could lead to earlier melanocyte death and removal, resulting in more extensive depigmentation in the presence of at least one *LP* and one *PATN1* allele [59].

## 8. Conclusions and Future Perspective

There have been huge advancements in equine genetics in recent years, and new white variants, coat color dilutions, and health-associated loci are being reported every year. However, there remain many poorly understood pigmentation traits requiring more research to uncover their genetic causes. For example, despite linkage of this phenotype to *KIT* in some breeds, the mechanisms underlying *Roan* are still poorly understood. *Rabicano* is characterized by white ticking on the flank, sometimes extending forward toward the barrel, and white banding on the tailhead, and has been linked to variants in possible regulatory regions affecting *KITLG* [62], but the genetic cause for *Rabicano* is still unknown. Few studies have focused on the health or reproduction of horses with multiple *KIT* variants. It has not been confirmed if homozygosity of multiple *Dominant White* alleles is truly lethal. More evidence on the deafness of overo and splashed white horses is needed to validate the anecdotal stories of deaf horses and to come to a conclusion about the cause of the hearing loss. It is important to continuously monitor the presence of *LP*, *PATN1*, and *LWO* variants in horse populations to mitigate adverse health effects. Studies are needed to elucidate the function of BLOC-2 and other *HPS* genes, which will help to improve the prognosis of *HPS*-related diseases in humans and reveal if horses suffer from the same defects due to *HPS* mutations. Understanding the etiology of equine coat color is crucial to improve strategies to breed healthier and more beautiful horses while controlling the allele frequencies of variants associated with adverse health effects.

## Figures and Tables

**Figure 1 animals-14-00451-f001:**
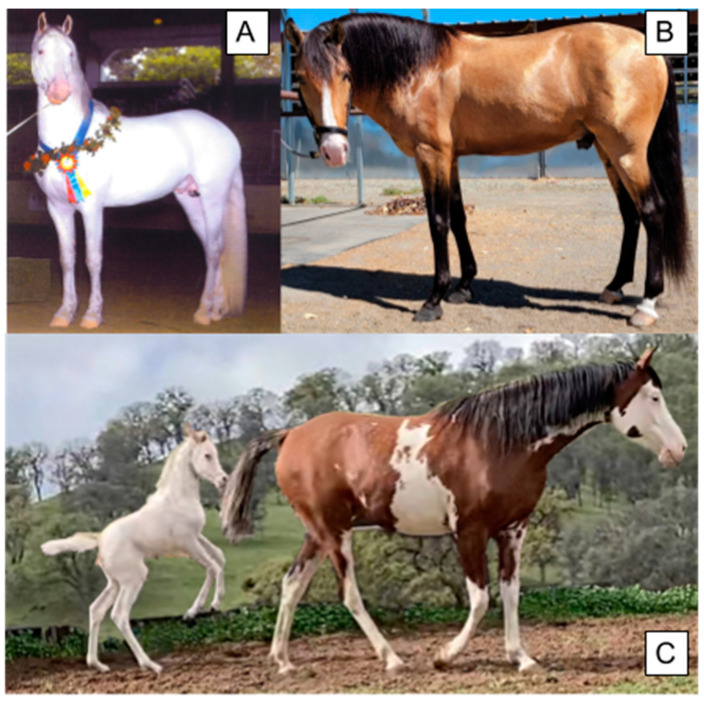
Various phenotypes caused by variants at the W locus, including full depigmentation, a sabino-like pattern, and minimal markings. Coat color genotypes and breeds are as follows: (**A**) *a/a E/e* (black) *W13/n*, Friesian-American White Horse cross. (**B**) *A/a E/e Cr/n* (Buckskin). (**C**) Foal—*A/A E/E* (bay) *W15/15*, Arabian, Dam—*A/A E/e* (bay) *W15/n*, Arabian.

**Figure 2 animals-14-00451-f002:**
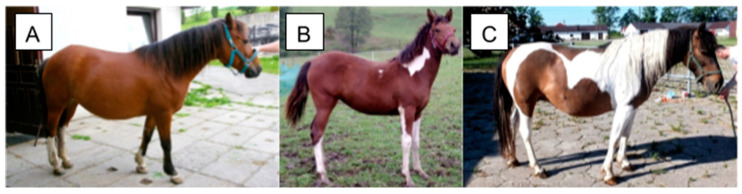
Varying degrees of the tobiano phenotype in the Hucul horse. The horse in (**A**) displays cryptic tobiano markings that can be confused with a solid horse with white markings despite carrying a TO variant. The horse in (**B**) shows minimal tobiano markings. The horse in (**C**) displays classic tobiano. All horses possess at least one TO variant. All horses display their base coat color as well as white markings. Figure adapted from [24]. Photographs by M. Pasternak.

**Figure 3 animals-14-00451-f003:**
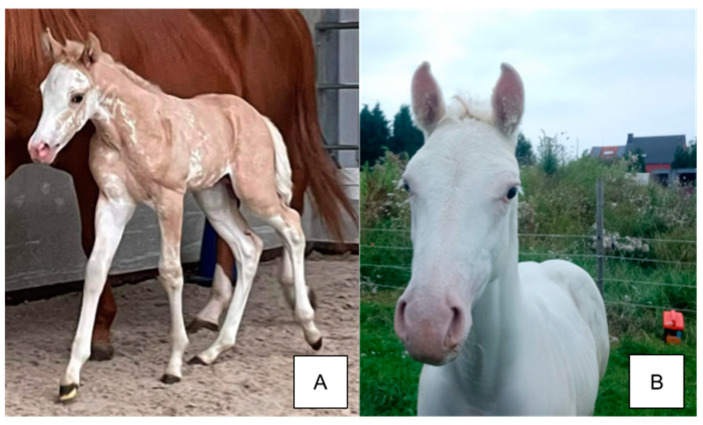
Incomplete dominance of the sabino trait. The American Saddlebred horse in (**A**) genotypes as SB1/sb1 and displays white socks, a blaze, and white spots stretching across the body while the Paint horse foal in (**B**) genotypes as SB1/SB1 and displays an all-white phenotype. Pink skin is exposed on the faces of both individuals.

**Figure 4 animals-14-00451-f004:**
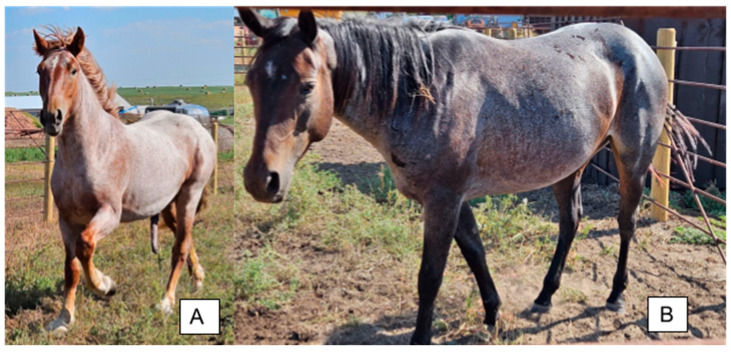
(**A**) Chestnut Roan Quarter horse with a mixture of white and chestnut hairs over the body (**B**) Bay Roan Quarter horse with a mixture of white and dark hairs over the body. The head and lower legs are solid in color for both horses.

**Figure 5 animals-14-00451-f005:**
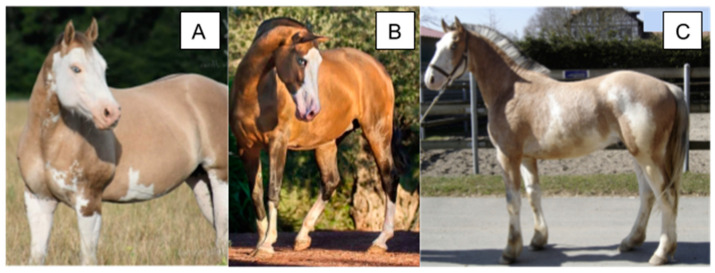
Common splashed white markings. (**A**) Paint horse with *SW7* displaying typical splashed markings on the face, legs, and body and blue eyes. This individual was reported as deaf and possesses one *Cream* variant diluting the coat color. Image adapted from [48]. (**B**) Pura Raza Española horse with *SW9* displaying blue eyes, four white limbs, and a white face. (**C**) Franches-Montagnes horse with the *Macchiato* variant, which causes a phenotype similar to splashed white and a coat color dilution similar to *Cream*. This individual was reported to not carry any coat dilutions other than *Macchiato*. Image adapted from [44].

**Figure 6 animals-14-00451-f006:**
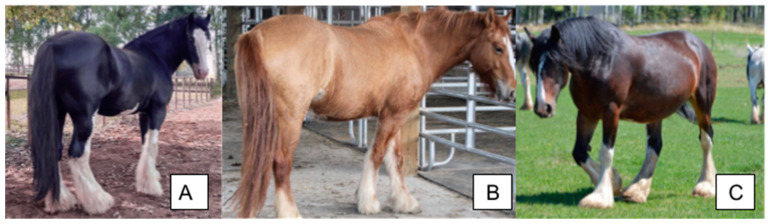
Phenotypes of horses with EDXW1-3. (**A**) Shire horse homozygous for both EDXW1 and EDXW3 displaying white spotting with white spots on the abdomen and blue eyes. (**B**) A Gypsy Vanner horse heterozygous for EDXW2 showing a similar, but less extensive, phenotype as Horse A. There is slight depigmentation below the back quarter. (**C**) Shire horse heterozygous for EDXW3 displaying white spotting on the abdomen, legs, and face. The horses pictured possess no white-causing variants except for the alleles explicitly mentioned.

**Figure 7 animals-14-00451-f007:**
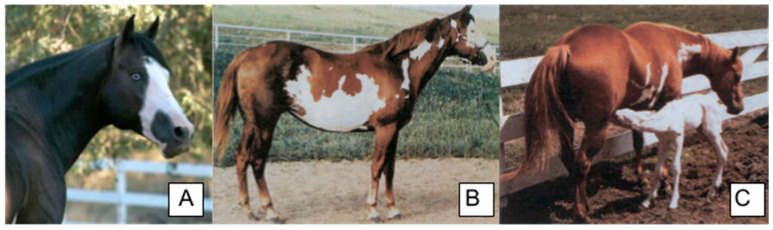
Common phenotypes of Paint horses with *Lethal White Overo (LWO)*. (**A**) *LWO/n* individual displays a blue eye and white blaze (photograph by M. Simmons). (**B**) Jagged body marks as a result of one copy of *LWO*. (**C**) All white homozygous *LWO* foal and a heterozygous *LWO* mare. The photographs in (**B**,**C**) were adapted from [57]).

**Figure 8 animals-14-00451-f008:**
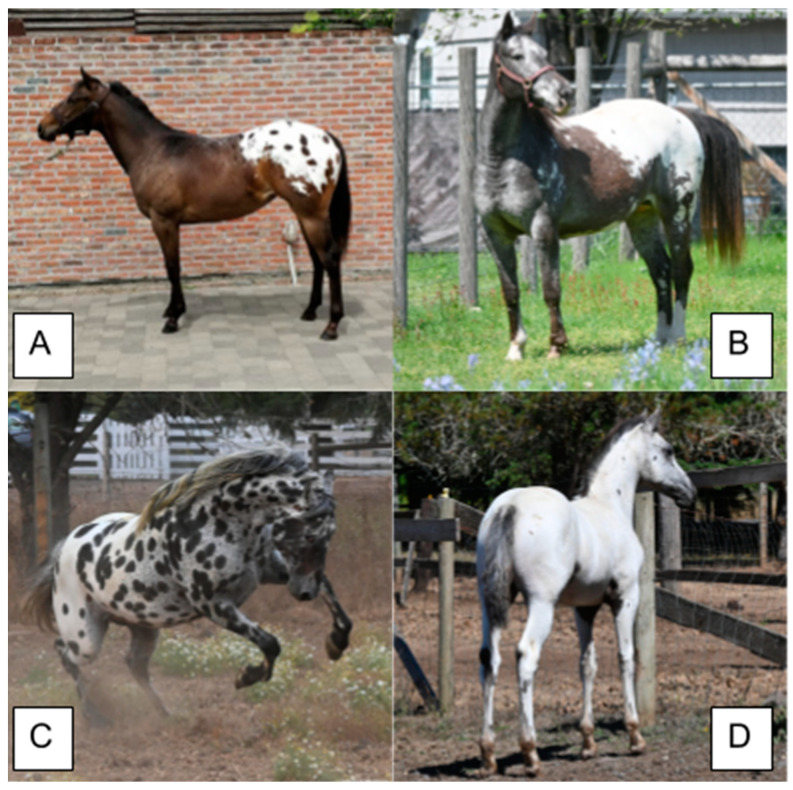
Phenotypes of Appaloosa horses attributed to *LP* and *PATN1*. (**A**) *LP/lp patn1/patn1* individual with a depigmented hind quarter. (**B**) *LP/LP patn1/patn1* individual with a depigmented flank and the few spot pattern. (**C**) *LP/lp PATN1/patn1* individual with full leopard spotting. (**D**) *LP/LP PATN1/PATN1* individual that is fully white with no other white spotting alleles.

**Table 1 animals-14-00451-t001:** White coat color traits and their distinguishing characteristics along with the associated genes and encoded proteins.

Trait	Allele Symbol	Gene/Region	Encoded Protein (Complex)	Distinguishing Characteristics (Phenotype)
*Dominant White*	*W*	*KIT*	KIT	White spots or white coat, pink skin (sabino-like, all white)
*Tobiano*	*TO*	*KIT*	KIT	Large white spots covering the body and crossing the spine, white legs, pink skin (tobiano)
*Sabino*	*SB1*	*KIT*	KIT	Jagged white markings or white coat, pink skin (sabino, all white)
*Roan*	*RN*	*KIT*	KIT	Interspersed white hairs distributed through the coat (roan)
*Splashed White*	*SW*	*MITF*, *PAX3*	MITF, PAX3	Deafness, blue eyes, smooth boarded white faces, abdomens, and legs (splashed white)
*Eden White*	*EDXW*	*HPS5*	HPS5 (BLOC-2)	White body spots, legs, and faces (sabino-like)
*Lethal White Overo*	*LWO*, *O*	*EDNRB*	EDNRB	White face, legs, and body spots that do not cross the spine (overo)
*Leopard Spotting*	*LP*	*TRPM1*	TRPM1	White hind quarter or white coat with colored spots, epistatic to *PATN1* (blanket or leopard)
*Pattern 1*	*PATN1*	*RFWD3*	RFWD3	Increases white spotting for individuals with a least one copy of *LP*

**Table 2 animals-14-00451-t002:** Genomic location, variant type, and phenotypes of *Dominant White KIT* variants.

Allele	Genomic Coordinate EquCab3.0	Type	Phenotype	Homozygotes	References
*W1*	chr3:79545942G>C	nonsense	All White	Not Observed	[2]
*W2*	chr3:79549540C>T	missense	All White	Not Observed	[2]
*W3*	chr3:79578535T>A	nonsense	All White	Not Observed	[2]
*W4*	chr3:79549780G>A	missense	All White	Not Observed	[2]
*W5*	chr3:79545900delC	small deletion	Sabino-like	Not Observed	[12]
*W6*	chr3:79573754C>T	missense	Sabino-like to All White	Not Observed	[12]
*W7*	chr3:79580000C>G	splice site	All White	Not Observed	[12]
*W8*	chr3:79545374C>T	splice site	Sabino-like	Not Observed	[12]
*W9*	chr3:79549797C>T	missense	All White	Not Observed	[12]
*W10*	chr3:79566925_79566928del	small deletion	Sabino-like to All White	Not Observed	[12]
*W11*	chr3:79540429C>A	splice site	All White	Not Observed	[12]
*W12*	chr3:79579755_79579779delAGACG	small deletion	Sabino-like	Not Observed	[17]
*W13*	chr3:79544066C>G	splice site	All White	Not Observed	[14]
*W14*	chr3:79544151_79544204del	gross deletion	All White	Not Observed	[14]
*W15*	chr3:79550351A>G	missense	Sabino-like to All White	Observed	[14,18]
*W16*	chr3:79540741T>A	missense	All White	Not Observed	[14]
*W17a*	chr3:79548265T>A	missense	All White	Not Observed	[14]
*W17b*	chr3:79548244A>G	missense	All White	Not Observed	[14]
*W18*	chr3:79553751C>T	splice site	Sabino-like	Not Observed	[15]
*W19*	chr3:79553776T>C	missense	Sabino-like	Observed	[15]
*W20*	chr3:7948220T>C	missense	No markings to Sabino-like	Observed	[15]
*W21*	chr3:79544174delG	small deletion	Sabino-like	Not Observed	[13]
*W22*	chr3:79548925_79550822del1898	gross deletion	Sabino-like	Not Observed	[11]
*W23*	chr3:79578484C>G	splice site	All White	Not Observed	[18]
*W24*	chr3:79545245C>T	splice site	All White	Not Observed	[10]
*W25*	chr3:77769878T>C	missense	All White	Not Observed	[16]
*W26*	chr3:79544150del	small deletion	Sabino-like	Not Observed	[16]
*W27*	chr3:79552028A>C	missense	All White	Not Observed	[16]
*W28*	chr3:79579925_79581197del	gross deletion	Sabino-like	Not Observed	[19]
*W30*	chr3:79548244T>A	missense	All White	Not Observed	[20]
*W31*	chr3:79618532_79618533insT	fs nonsense	Sabino-like	Not Observed	[7]
*W32*	chr3:79538738C>T	missense	No markings to Sabino-like	Observed	[7]
*W33*	chr3:79545248T>A	missense	Sabino-like	Not Observed	[5]
*W34*	chr3:79566881T>C	missense	No markings to Sabino-like	Observed	[8]
*W35*	chr3:79618649A>C	UTR variant	No markings to Sabino-like	Observed	[6]

**Table 3 animals-14-00451-t003:** Genomic coordinates, gene, variant type, and phenotypes of *Tobiano* and *Sabino* variants and the *Roan*-associated SNP.

Allele	Gene	Genome Coordinate EquCab3.0	Type	Phenotype	Reference
*SB1*	*KIT*	chr3:79544206A>T	splice site	White markings with jagged borders and pink skin. Homozygotes are all white with pink skin	[9]
*TO*	Intergenic	chr3:42737476_79471993inv	paracentric inversion	Large white areas across the body, pink skin	[36]
*RN*	*KIT*	chr3:79543439A>G	associated SNP	White hairs distributed throughout the coat	[37]

**Table 4 animals-14-00451-t004:** Genomic coordinates, gene, variant type, and phenotypes of *Splashed White* variants.

Allele	Gene	Genomic Coordinate EquCab3.0	Type	Phenotype	Reference
*SW1*	*MITF*	chr16:21579201delinsATAATAACCTA	small indel	Splashed Markings, Deafness, Blue Eyes	[44]
*SW2*	*PAX3*	chr6:11199026C>T	missense	Minimal Splashed Markings, Deafness, Blue Eyes	[44]
*SW3*	*MITF*	chr16:21567245_21567249del	small deletion	Splashed Markings, Deafness, Blue Eyes	[44]
*SW4*	*PAX3*	chr6:11199140G>C	missense	Minimal Splashed Markings, Blue Eyes	[15]
*SW5*	*MITF*	chr16:21503211_21566617del	gross deletion	Splashed Markings, Deafness, Blue Eyes	[45]
*SW6*	*MITF*	chr16:21551060_21559770del	gross deletion	Splashed Markings, Deafness, Blue Eyes	[46]
*SW7*	*MITF*	chr16:21559953_21559955del	small deletion	Splashed Markings, Deafness, Blue Eyes	[48]
*SW8*	*MITF*	chr16:21555811_21558139del	gross deletion	Splashed Markings, Deafness, Blue Eyes	[43]
*SW9*	*MITF*	chr16:21559940A>T	missense	Splashed Markings, Blue Eyes	[47]
*SW10*	*PAX3*	chr6:11196181C>T	nonsense	Splashed Markings, Blue Eyes	[47]

**Table 5 animals-14-00451-t005:** Genomic coordinates, variant type, and phenotypes of Eden White variants in HPS5.

Allele	Genomic Coordinate EquCab3.0	Type	Phenotype	Reference
*EDX* *W1*	chr7:88741616T>A	missense	High socks, belly spots, blazes	[56]
*EDX* *W2*	chr7:88751667G>A	missense	High socks, belly spots, blazes	[56]
*EDX* *W3*	chr7:88766673T>C	splice site	High socks, belly spots, blazes	[56]

**Table 6 animals-14-00451-t006:** Genomic coordinates, genes, variant type, and phenotypes of *LP* and *PATN1*.

Allele	Gene	Genomic Coordinate EquCab3.0	Type	Phenotype	Reference
*LP*	*TRPM1*	chr1:108297929_108297930insN[1378]	insertion	White hind quarter, all white (hom), leopard spots	[27]
*PATN1*	*RFWD3*	chr3:24352525T>G	UTR variant	Increases white and decreases number of spots only for *LP/LP* or *LP/lp* individuals	[59]

## Data Availability

Not applicable.

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
