# Peer review of "Spotting the Pattern: A Review on White Coat Color in the Domestic Horse"

_animals, 2024, doi:10.3390/ani14030451_

Round 1

Reviewer 1 Report

Comments and Suggestions for Authors

Author Response

  1. Added a list of genes in the introduction associated with white coats
  2. Split this sentence into two sentences and edited for clarity
  3. Removed the 2nd and 3rd appearances "(indel, fs, etc)"
  4. Edited this sentence to remove redundancy 
  5. Cited Stachurska and Jason (2015) in the Tobiano section and added the corresponding reference
  6. We are unaware what other equine research regarding the KIT region that you refer to.  We mention the active research in the KIT region regarding Roan

Reviewer 2 Report

Comments and Suggestions for Authors

The manuscript is well written profound review focusing on different white color patterns in the domestic horse. The review-article based on 60 references published in 1969-2023. Recent data used in tables characterizing variety of phenotypes and genomic coordinates are mostly from 2013-2023, hence combined with earlier knowledge concerning horses’ coat patterns and health risks in the manuscript text parts. There are also two unpublished manuscripts by the authors (submitted 2023) included.

The phenotypic descriptions are detailed, however, it seems to the referee, that authors should provide an additional paragraph or table in intro with the list and short distinguishing characteristics of reviewed coat patterns (phenotypes with used abbreviations and alleles) as not all readers are familiar to the specific sabino, leopard, and etc. horses.

Specific comments/suggestions:

Nomenclature. Does the difference in the use of some capital/ lower-case letter have specific meaning? Please unify if any (for example recessive allele symbols n and N (line 461); unknown allele in heterozygous genotypes x and X (line 604, line 606 also, "x" should explained in the text).
It is not clear if the genes are written in italic and corresponding proteins using regular font pervade the text.

Tables. Title should give information about entire data in the table. Please correct.
E g. Table 1: chromosomal location (ECA3) must be removed from the title as it is given in the genomic coordinates.
Please correct the column captions “Coordinates” (it could be changed to “Genomic coordinate” and add reference sequence, please), correct column “Homozygotes” that gives information of the homozygote occurrence regarding the allele (“not observed” versus “viable” change to “observed”), correct caption “Pigment Phenotype”. What authors mean by the term Pigment Phenotype?
Elsewhere, the title and caption of the column with mutation site coordinates should be corrected.

Figures. Please unify and correct references to the photos.
Figure 3, 4, 6. Please replace the thank you to the (anonymous?) horse owners of the photographs into Acknowledgments part.

References and citing.
OMIA should have a reference (line 94 in the manuscript).
Reference 16 (line 708) is not found in the text.
Please insert citation for the Table 5 into text. 

Author Response

Thank you for taking the time to review the manuscript and for providing specific instructions on how we could improve. We felt that all your comments were valid and needed edits. In response to your comments, we edited the following:

  1. Added a small table in the intro summarizing the traits described, the differing characteristics, the gene name, the protein name, and the symbol used for the trait.
  2. In response to nomenclature, we... 
    1. unified use of "n" to only lowercase
    2. unified use of "X" to uppercase; explained the meaning of "X" in the text
    3. clarified protein and gene nomenclature in the new table in the intro (see 1), fixed phrases that did not agree with this nomenclature
  3. To unify tables we... 
    1. Removed ECAX from titles and added to titles to reference all data in the data
    2. Changed coordinates to genomic coordinates and added reference sequence EquCab3.0 for all tables
    3. Corrected the Homozygotes column to read "Observed" instead of "viable"
    4. Changed pigment phenotype to phenotype
    5. Fixed other tables to unify
  4. Figures 
    1. Unified in-text figure references
    2. Placed "thank yous" in the acknowledgments
  5. References/Citing
    1. Cited OMIA 
    2. Added reference 16 in section 2 paragraph 1
    3. Ensured all figures and tables were cited in the text